# Diffraction from Nanocrystal Superlattices

**DOI:** 10.3390/nano12101781

**Published:** 2022-05-23

**Authors:** Antonio Cervellino, Ruggero Frison

**Affiliations:** 1Swiss Light Source, Paul Scherrer Institut, 5232 Villigen, Switzerland; 2Physik-Institut, Universität Zürich, Winterthurerstrasse 190, 8057 Zurich, Switzerland; ruggero.frison@physik.uzh.ch

**Keywords:** superlattices, nanocrystals, X-ray diffraction

## Abstract

Diffraction from a lattice of periodically spaced crystals is a topic of current interest because of the great development of self-organised superlattices (SL) of nanocrystals (NC). The self-organisation of NC into SL has theoretical interest, but especially a rich application prospect, as the coherent organisation has large effects on a wide range of material properties. Diffraction is a key method to understand the type and quality of SL ordering. Hereby, the characteristic diffraction signature of an SL of NC—together with the characteristic types of disorder—are theoretically explored.

## 1. Introduction

We will explore the diffraction characteristics of supercrystals (SCs) as superlattices (SLs) of nanocrystals (NCs) where the periodic entity is itself a small crystal. There is a widespread current interest on such materials, driven by the changes in properties that the periodic organisation of NCs into an SC yields, mainly for applications in optics [1], electionics [2], and catalysis [3]. The synthesis of SCs is a result of the always more sophisticated ways of synthesising NCs (and nanoparticles in general) with very sharp distributions in size and well defined faceted shape [4,5]. These NCs then, under proper conditions, self-organise, forming an SL and thence an SC [6,7]. A recent brief current perspective on SCs and their properties can be found in [8].

With the advances in SL synthesis methods, characterization tools such as small and wide-angle scattering methods that allow for probing the SLs in their real sample environments and, possibly, in real time, become increasingly important. X-ray scattering experiments allow for obtaining structural information on the ordering state of the assembly along with their lattice parameters and symmetry. A general and comprehensive review of the application of the small-angle scattering techniques has been given by Li et al. [9], and more recently by Jiang et al. [10], which includes a detailed discussion of the latest developments like GISAXS or coherent-SAXS. The analysis of 1D SAXS data is well-established [11]. Senesi et al. have discussed the various approximations which can be introduced in the structure factor calculation to describe the contributions of the particles’ polydispersity, orientational disorder, and position [12]. The analysis of 2D SAXS or GISAXS data, although in general performed at a more qualitative level than in the 1D case, has become increasingly popular as the 2D images collected during the experiment provide an overview of the reciprocal space which allow for an easier evaluation of the symmetry and coherence lengths of the assembly [13,14].

The use of wide-angle X-ray scattering data (WAXS) is far less common in the study of SLs, although it has been shown that information about structural coherence along two and three directions, degree of orientational order, and coherent sizes can be retrieved [15].

Here, we deepen the discussion of diffraction theory of SCs. In particular, we will focus on the diffraction signature of imperfectly ordered SCs, especially concerning the effect of NCs of different sizes in the SL nodes, and the effect of slight rotation of the component NCs with respect to each other. We derive expressions for the powder diffraction total scattering intensity, which includes the size dispersion effect and allow a faster evaluation of Debye Scattering Equation. The orientational disorder will be developed only partly in this paper, delegating the full discussion to an upcoming technical paper. The results presented here will be useful for the quantitative interpretation of diffraction data from SLs.

## 2. Materials and Methods

The numerical simulations of powder diffraction patterns hereby presented were computed using the DEBUSSY software suite [16] and ad hoc written code in Fortran2008 (available by email from the authors) to include the analytic expressions derived here. The simulations of single crystal diffraction patterns were computed using the ZODS program [17]. The programs build model crystals by means of Monte Carlo (MC) simulations and compute their Fourier transforms according to the standard formula for kinematic scattering.

## 3. Perfect Superlattice of Identical Nanocrystals

A perfect superlattice of identical nanocrystals, perfectly equioriented in space and periodically arranged without any defect, can be dealt with as a conventional crystalline structure with a large unit cell. Several papers deal with diffraction from SCs, especially within some experimental framework, with a comprehensive review in Jiang et al. [10] including sophisticated techniques as GISAXS and coherent SAXS. However, there is also an interesting and simple analytic formula describing the diffraction amplitude of such supercrystal, if some inessential shape restrictions are assumed. We will consider parallelohedral nanocrystals, extended along the unit cell vectors $\underline{a}$, $\underline{b}$, $\underline{c}$, and whose nanocrystal lattice coordinates are defined by integers $n_{a},n_{b},n_{c}$:(1)$$\left(n_{a}\underline{a},n_{b}\underline{b},n_{c}\underline{c}\right)\;|\;0⩽n_{a}⩽N_{a},\;0⩽n_{b}⩽N_{b},\;0⩽n_{c}⩽N_{c}.$$The superlattice cell vectors we suppose to be direct multiples of the crystal cell vectors:(2)$$\begin{array}{ccc} {\underline{a}}_{S} & = & \left(N_{a}+s_{a}\right)\underline{a};\\ {\underline{b}}_{S} & = & \left(N_{b}+s_{b}\right)\underline{b};\\ {\underline{c}}_{S} & = & \left(N_{c}+s_{c}\right)\underline{c}. \end{array}$$The spacings constants $s_{a},s_{b},s_{c}$ are supposed to be positive; otherwise, we would have coalescence (or even overlap) of the crystal domains. Coalescence would bring us to polycrystalline matter, which is quite another issue. Instead, the spacings are supposed to be filled by some kind of ligand. Again, we assume a parallelohedral shape for the supercrystals, assuming that the occupied superlattice nodes lie at integer multiples of ${\underline{a}}_{S},{\underline{b}}_{S},{\underline{c}}_{S}$, similarly to Equation (Equation 1):(3)$$\left(m_{a}{\underline{a}}_{S},m_{b}{\underline{b}}_{S},m_{c}{\underline{c}}_{S}\right)\;|\;0⩽m_{a}⩽M_{a},\;0⩽m_{b}⩽M_{b},\;0⩽m_{c}⩽M_{c}.$$

We abstain in the following from describing the atomic content of the unit cell; we will assume that each NC unit cell contains just one point scatterer of unit scattering power in the origin. Generalisation to real NCs with a specified unit cell content is straightforward but able to unnecessarily complicate the notation. The NC’s scattering density is described formally in Equation (Equation 4).

The unit cell of the superlattice contains instead a single NC. We can arbitrarily set each superlattice node in the NC’s scattering barycentrum
$$\underline{C}=\left(\left(N_{a}+1\right)\underline{a},\left(N_{b}+1\right)\underline{b},\left(N_{c}+1\right)\underline{c}\right)/2.$$As such, the scattering density of an SC is just that of the SL (with—again—unit power point scatterers on the lattice nodes, see Equation (5)) convoluted with the scattering density of an NC (Equation (6)): (4)$$\begin{array}{ccc} \rho_{NC}\left(\underline{r}\right) & = & \sum_{n_{a}=1}^{N_{a}}\sum_{n_{b}=1}^{N_{b}}\sum_{n_{c}=1}^{N_{c}}\delta \left(\underline{r}-\left(n_{a}\underline{a},n_{b}\underline{b},n_{c}\underline{c}\right)+\underline{C}\right); \end{array}$$(5)$$\begin{array}{ccc} \rho_{SL}\left(\underline{r}\right) & = & \sum_{m_{a}=1}^{M_{a}}\sum_{m_{b}=1}^{M_{b}}\sum_{m_{c}=1}^{M_{c}}\delta \left(\underline{r}-\left(m_{a}{\underline{a}}_{S},m_{b}{\underline{b}}_{S},m_{c}{\underline{c}}_{S}\right)\right); \end{array}$$(6)$$\begin{array}{cc} \rho_{SC}\left(\underline{r}\right) & =\int_{R^{3}}d^{3}{\underline{r}}^{\prime }\;\rho_{NC}\left({\underline{r}}^{\prime }\right)\rho_{SL}\left(\underline{r}-{\underline{r}}^{\prime }\right)\\ & =\;\;\;\sum_{m_{a}=1}^{M_{a}}\sum_{m_{b}=1}^{M_{b}}\sum_{m_{c}=1}^{M_{c}}\sum_{n_{a}=1}^{N_{a}}\sum_{n_{b}=1}^{N_{b}}\sum_{n_{c}=1}^{N_{c}}\;\;\delta \left(\underline{r}-\;\left(n_{a}\underline{a},n_{b}\underline{b},n_{c}\underline{c}\right)\;+\;\underline{C}\;-\;\left(m_{a}{\underline{a}}_{S},m_{b}{\underline{b}}_{S},m_{c}{\underline{c}}_{S}\right)\;\right) \end{array}$$It follows that the SC’s scattering amplitude (the Fourier transform) is the product of the scattering amplitude of NC times that of an SL decorated with unit point scatterers: (7)$$\begin{array}{ccc} F_{NC}\left(\underline{q}\right) & = & \int_{R^{3}}d^{3}\underline{r}\;\rho_{NC}\left(\underline{r}\right)e^{2\pi i\underline{q}\cdot \underline{r}}\;; \end{array}$$(8)$$\begin{array}{ccc} F_{SL}\left(\underline{q}\right) & = & \int_{R^{3}}d^{3}\underline{r}\;\rho_{SL}\left(\underline{r}\right)e^{2\pi i\underline{q}\cdot \underline{r}}\;; \end{array}$$(9)$$\begin{array}{ccc} F_{SC}\left(\underline{q}\right) & = & \int_{R^{3}}d^{3}\underline{r}\;\rho_{SC}\left(\underline{r}\right)e^{2\pi i\underline{q}\cdot \underline{r}}=F_{NC}\left(\underline{q}\right)F_{SL}\left(\underline{q}\right) \end{array}$$Here, $\underline{q}$ is the transferred momentum vector, whose length is $q=\left|\underline{q}\right|=2\sin\left(\theta \right)/\lambda$, with λ the incident wavelength and θ half of the deflection angle. The transform in Equation (Equation 7) has been historically evaluated by Max von Laue [18,19], as
(10)$$F_{NC}\left(\underline{q}\right)=\frac{\sin\left(N_{a}\pi \underline{q}\cdot \underline{a}\right)}{\sin\left(\pi \underline{q}\cdot \underline{a}\right)}\frac{\sin\left(N_{b}\pi \underline{q}\cdot \underline{b}\right)}{\sin\left(\pi \underline{q}\cdot \underline{b}\right)}\frac{\sin\left(N_{c}\pi \underline{q}\cdot \underline{c}\right)}{\sin\left(\pi \underline{q}\cdot \underline{c}\right)}$$In more modern form, using the Chebyshev polynomials of the second kind $U_{k}\left(x\right)$ (see [20], Equation (Equation 22)), we can rewrite it as
(11)$$F_{NC}\left(\underline{q}\right)=U_{N_{a}-1}\left(\cos\left(\pi \underline{q}\cdot \underline{a}\right)\right)U_{N_{b}-1}\left(\cos\left(\pi \underline{q}\cdot \underline{b}\right)\right)U_{N_{c}-1}\left(\cos\left(\pi \underline{q}\cdot \underline{c}\right)\right)$$Similarly,
(12)$$F_{SL}\left(\underline{q}\right)=e^{-2\pi i\underline{q}\cdot {\underline{C}}_{S}}U_{M_{a}-1}\left(\cos\left(\pi \underline{q}\cdot \underline{a_{S}}\right)\right)U_{M_{b}-1}\left(\cos\left(\pi \underline{q}\cdot \underline{b_{S}}\right)\right)U_{M_{c}-1}\left(\cos\left(\pi \underline{q}\cdot \underline{c_{S}}\right)\right)$$The phase factor is because we have not referred our SL slab to its scattering barycentrum ${\underline{C}}_{S}=\left(\left(M_{a}+1\right){\underline{a}}_{S},\left(M_{b}+1\right){\underline{b}}_{S},\left(M_{c}+1\right){\underline{c}}_{S}\right)/2$, but it is inessential. In fact, to obtain the scattered intensity $I_{SC}\left(\underline{q}\right)$, we take the square modulus of $F_{SC}\left(\underline{q}\right)$,
(13)$$I_{SC}\left(\underline{q}\right)={\left|F_{NC}\left(\underline{q}\right)F_{SL}\left(\underline{q}\right)\right|}^{2}=F_{NC}^{2}\left(\underline{q}\right){\left|F_{SL}\left(\underline{q}\right)\right|}^{2}$$
where the phase factor disappears, and we have a product of six squared Chebyshev polynomials. A simple graph shows these simple functions for $\underline{q}=h{\underline{a}}^{*}$, along the NC reciprocal axis ${\underline{a}}^{*}$. The reciprocal space vectors are defined by
$${\underline{a}}^{*}\cdot \underline{a}={\underline{b}}^{*}\cdot \underline{b}={\underline{c}}^{*}\cdot \underline{c}=1;\;{\underline{a}}^{*}\cdot \underline{b}={\underline{b}}^{*}\cdot \underline{c}={\underline{c}}^{*}\cdot \underline{a}={\underline{a}}^{*}\cdot \underline{c}={\underline{b}}^{*}\cdot \underline{a}={\underline{c}}^{*}\cdot \underline{b}=0.$$We also assume—for this example—that the SL vectors
$${\underline{a}}_{S}\propto \underline{a};\;{\underline{b}}_{S}\propto \underline{b};\;{\underline{c}}_{S}\propto \underline{c}.$$Therefore, if $\underline{q}=h{\underline{a}}^{*}$, then
$$\cos\left(\pi \underline{q}\cdot \underline{b}\right)=\cos\left(\pi \underline{q}\cdot \underline{c}\right)\cos\left(\pi \underline{q}\cdot \underline{b_{S}}\right)=\cos\left(\pi \underline{q}\cdot \underline{c_{S}}\right)=\cos\left(0\right)=1$$
and
$$U_{N_{b}-1}\left(1\right)=N_{b};\;U_{N_{c}-1}\left(1\right)=N_{c};\;U_{M_{b}-1}\left(1\right)=M_{b};\;U_{M_{c}-1}\left(1\right)=M_{c}.$$We can omit these constant factors without prejudice. The SC intensity along $\underline{q}=h{\underline{a}}^{*}$ is then just
$$I_{SC}\left(h\right)=I_{NC}\left(h\right)I_{SL}\left(h\right)=U_{N_{a}-1}^{2}\left(\cos\left(\pi h\right)\right)U_{M_{a}-1}^{2}\left(\cos\left(\pi h(a_{S}/a)\right)\right)$$In Figure 1, we plot both $I_{NC}\left(h\right)=U_{N_{a}-1}^{2}\left(\cos\left(\pi h\right)\right)$ and $I_{SC}\left(h\right)/M_{a}^{2}$; the last scaling sets $0<I_{SL}\left(h\right)<1$ for convenience.

## 4. Superlattice of Not Identical Objects

In this section, we explore the case when the NCs arranged on the SL are not all equal sized. We will call this situation the size disorder effect (SDE).

We still assume a paralleloidal shape. The dimensional constants $N_{a},N_{b},N_{c}$ and the spacing constants $s_{a},s_{b},s_{c}$ (see Equation (Equation 2)) may sometimes not be indeed constant and immutable throughout the structure. We will suppose instead that all of them may be statistically described by (narrow) distributions over the positive real axis, each having a defined average and variance and all finite superior moments. The simplest and most widely used such distributions are lognormals. Thus, for instance, we suppose that $N_{a}$ has average $\langle N_{a}\rangle$ and variance $V_{N_{a}}$. We also introduce for convenience the fractional dispersions
$$\xi \equiv \frac{\sqrt{V_{N_{a}}}}{\langle N_{a}\rangle };\;\tau \equiv \frac{\sqrt{V_{s_{a}}}}{\langle s_{a}\rangle }$$A lognormal probability density describing $N_{a}$ (represented by the continuous variable *X*) is
(14)$$P_{N_{a}}\left(X\right)=\frac{1}{X\sqrt{2\pi \;\log\left({\langle N_{a}\rangle }^{2}\left(1+\xi^{2}\right)\right)}}\exp\left[-\frac{1}{2}\frac{{\left(\log\left(X\right)-\log\left(\langle N_{a}\rangle \right)+\left(1/2\right)\log\left(1+\xi^{2}\right)\right)}^{2}}{\log\left(1+\xi^{2}\right)}\right]$$In addition, similarly, for $s_{a}$, with associated variable *Y*,
(15)$$P_{s_{a}}\left(Y\right)=\frac{1}{Y\sqrt{2\pi \;\log\left({\langle s_{a}\rangle }^{2}\left(1+\tau^{2}\right)\right)}}\exp\left[-\frac{1}{2}\frac{{\left(\log\left(Y\right)-\log\left(\langle s_{a}\rangle \right)+\left(1/2\right)\log\left(1+\tau^{2}\right)\right)}^{2}}{\log\left(1+\tau^{2}\right)}\right]$$

Similarly, for $N_{b}$, $N_{c}$ and $s_{b}$, $s_{c}$, all the averages are straightly denoted
$$\langle N_{a}\rangle ,\;\langle N_{b}\rangle ,\;\langle N_{c}\rangle ,\;\langle s_{a}\rangle ,\;\langle s_{b}\rangle ,\;\langle c_{c}\rangle ;$$
and the variances
$$V_{N_{a}},\;V_{N_{b}},\;V_{N_{c}},\;V_{s_{a}},\;V_{s_{b}},\;V_{s_{c}}.$$

### 4.1. 1D Superlattice

This is a simple set of nanocrystals on a line, hence forming a rod. Notation: if a variable *X* is distributed according to a given probability density P(X) whose moments are all finite, we denote $\langle X\rangle$ its average (normalized first moment) and $V_{X}$ its variance (normalized central moment) under P(X).

Consider first the fully ordered case, where
(16)$$\begin{array}{ccc} \langle N_{a}\rangle & = & N_{a},\;V_{N_{a}}=0;\;\langle s_{a}\rangle =s_{a},\;V_{s_{a}}=0. \end{array}$$

Firstly, for a finite sequence of equispaced points of length $M_{a}$, the multiplicity of the zero distance is $M_{a}$, while that of any pair of *k*-spaced nodes is
(17)$$\mu_{d_{k}}=2\left(M_{a}-\left|k\right|\right)$$

The average distance between two nodes spaced by *k* superlattice sites will be simply
$$d_{k}=k\left(N_{a}+s_{a}\right)a.$$If now we remove the assumptions in Equation (Equation 16), we have to average over the distribution of *every* variable segment. Supposing now every segment is variable, thus
$$\begin{array}{ccc} \langle d_{k}\rangle & = & a\int dX_{1}\;\int dX_{2}\;\ldots \int dX_{k}\;\;\int dY_{1}\;\int dY_{2}\;\ldots \int dY_{k}\;\left(\sum_{\ell =1}^{k}X_{\ell }+\sum_{\ell =1}^{k}Y_{\ell }\right)\times \end{array}$$
(18)$$\begin{array}{ccc} & \times & P_{N_{a}}\left(X_{1}\right)P_{N_{a}}\left(X_{2}\right)\ldots P_{N_{a}}\left(X_{k}\right)\;P_{s_{a}}\left(Y_{1}\right)P_{s_{a}}\left(Y_{2}\right)\ldots P_{s_{a}}\left(Y_{k}\right) \end{array}$$
(19)$$\begin{array}{ccc} & = & a\sum_{\ell =1}^{k}\int dX_{\ell }\;\;X_{\ell }\;P_{N_{a}}\left(X_{\ell }\right)+a\sum_{\ell =1}^{k}\int dY_{\ell }\;\;Y_{\ell }\;P_{S_{a}}\left(Y_{\ell }\right) \end{array}$$
[because all P_*_ functions are normalised to 1]
(20)$$\begin{array}{ccc} & = & k\left(\langle N_{a}\rangle +\langle s_{a}\rangle \right)a\;\mathrm{by}\;\mathrm{definition} \end{array}$$Similarly, for the variance, repeating similar passages, we obtain
(21)$$V_{d_{k}}=\left|k\right|\left(V_{N_{a}}+V_{s_{a}}\right)a^{2}$$It is clear that the effect on the interatomic distances of the variability of the size $N_{a}$ and that of the spacing $s_{a}$ are indistinguishable. We will consider—unless otherwise specified—a single parameter $\eta_{a}\equiv N_{a}+s_{a}$, so that
(22)$$\langle d_{k}\rangle =k\;\langle \eta_{a}\rangle a;\;V_{d_{k}}=\left|k\right|\;V_{\eta_{a}}a^{2}$$

### 4.2. 2D and 3D Superlattices

The fully ordered case is described in Section 3. We will hereby only consider the orthorhombic case where
$$\underline{a}\cdot \underline{b}=\underline{b}\cdot \underline{c}=\underline{c}\cdot \underline{a}=0;\;{\underline{a}}_{S}=\eta_{a}\underline{a},\;{\underline{b}}_{S}=\eta_{a}\underline{b},\;{\underline{c}}_{S}=\eta_{a}\underline{c},$$
where, as before,
$$\eta_{a}\equiv N_{a}+s_{a};\;\eta_{b}\equiv N_{b}+s_{b};\;\eta_{c}\equiv N_{c}+s_{c}.$$

In the ordered case,
(23)$$\langle \eta_{a}\rangle =N_{a},\;V_{\eta_{a}}=0;\;\langle \eta_{b}\rangle =N_{b},\;V_{\eta_{b}}=0;\;\langle \eta_{c}\rangle =N_{c},\;V_{\eta_{c}}=0.$$

Consider an SL formed by a parallelogram
$$m_{a}=1,\ldots ,M_{a};\;m_{b}=1,\ldots ,M_{b};\;m_{c}=1,\ldots ,M_{c}.$$The vector distance between two SL nodes $\underline{M}=\left(m_{a},m_{b},m_{c}\right)$ and ${\underline{M}}^{\prime }=\left(m_{a}^{\prime },m_{b}^{\prime },m_{c}^{\prime }\right)$ spaced by $\underline{K}\equiv \left(k_{a},k_{b},k_{c}\right)=\left(m_{a}^{\prime },m_{b}^{\prime },m_{c}^{\prime }\right)-\left(m_{a},m_{b},m_{c}\right)$ will be
$${\underline{d}}_{\underline{K}}=k_{a}{\underline{a}}_{S}+k_{b}{\underline{b}}_{S}+k_{c}{\underline{c}}_{S}=\left(k_{a}\eta_{a}a,k_{b}\eta_{b}b,k_{c}\eta_{c}c\right)$$In addition, it is immediate to generalise Equation (Equation 17) for the multiplicity of ${\underline{d}}_{\underline{K}}$ as
(24)$$\mu_{{\underline{d}}_{\underline{K}}}=\left(M_{a}-\left|k_{a}\right|\right)\left(M_{b}-\left|k_{b}\right|\right)\left(M_{c}-\left|k_{c}\right|\right).$$

The total distance between two point scatterers belonging each to one of the two NC centered at the $\underline{M}$ and ${\underline{M}}^{\prime }$ SL nodes must also take into account the difference between respective position vectors $\underline{n}=\left(n_{a},n_{b},n_{c}\right)$ and ${\underline{n}}^{\prime }=\left(n_{a}^{\prime },n_{b}^{\prime },n_{c}^{\prime }\right)$ in the generic NC lattice $\underline{g}\equiv \underline{n}-{\underline{n}}^{\prime }$. It results in
(25)$${\underline{d}}_{\underline{K},\underline{g}}={\underline{d}}_{\underline{K}}+{\underline{d}}_{\underline{g}}={\underline{d}}_{\underline{K}}+g_{a}\underline{a}+g_{b}\underline{b}+g_{c}\underline{c}=\left(k_{a}\eta_{a}a,k_{b}\eta_{b}b,k_{c}\eta_{c}c\right)+\left(g_{a}a,g_{b}b,g_{c}c\right)$$

If we consider instead the η parameters to follow a probability density with all finite moments, we can repeat the calculations in Section 4.1 component by component. We have to add an assumption—that the joint distribution is the product of the single variable distributions, or
$$P_{\eta_{a},\eta_{b},\eta_{c}}\left(X_{a},X_{b},X_{c}\right)=P_{\eta_{a}}\left(X_{a}\right)P_{\eta_{b}}\left(X_{b}\right)P_{\eta_{c}}\left(X_{c}\right)$$This will cause the covariance to be diagonal. Removing this assumption is straightforward, but it leads to far more complex bookkeeping.

We then have the vector average
(26)$$\langle {\underline{d}}_{\underline{K}}\rangle =\left(k_{a}\langle \eta_{a}\rangle a,k_{b}\langle \eta_{b}\rangle b,k_{c}\langle \eta_{c}\rangle c\right)$$The NC-related distance vector ${\underline{d}}_{\underline{g}}$ in Equation (Equation 25) is constant; therefore, it adds to the average and does not contribute to the variance. The averages result in being
(27)$$\langle {\underline{d}}_{\underline{K},\underline{g}}\rangle =\left(\left(k_{a}\langle \eta_{a}\rangle +g_{a}\right)a,\left(k_{b}\langle \eta_{b}\rangle +g_{b}\right)b,\left(k_{c}\langle \eta_{c}\rangle +g_{c}\right)c\right).$$In addition, we have a diagonal covariance matrix $V_{{\underline{d}}_{\underline{K},\underline{g}}}$ that is actually independent from $\underline{g}$:(28)$$V_{{\underline{d}}_{\underline{K},\underline{g}}}=\left(\begin{array}{ccc} \left|k_{a}\right|V_{\eta_{a}} & 0 & 0\\ 0 & \left|k_{b}\right|V_{\eta_{b}} & 0\\ 0 & 0 & \left|k_{c}\right|V_{\eta_{c}} \end{array}\right)$$

We cannot be too specific on the form of the 3D distribution of ${\underline{d}}_{\underline{K}}$; however, it is not wrong to assume it being a 3D Gaussian with specified averages and covariance matrix. Then, we would have
(29)$$P\left({\underline{d}}_{\underline{K},\underline{g}}\right)=\frac{1}{{\left(2\pi \right)}^{3/2}\sqrt{\det V_{{\underline{d}}_{\underline{K},\underline{g}}}}}\exp\left[-\frac{1}{2}\left({\underline{d}}_{\underline{K},\underline{g}}-\langle {\underline{d}}_{\underline{K},\underline{g}}\rangle \right)\cdot V_{{\underline{d}}_{\underline{K},\underline{g}}}\left({\underline{d}}_{\underline{K},\underline{g}}-\langle {\underline{d}}_{\underline{K},\underline{g}}\rangle \right)\right]$$

### 4.3. Powder Diffraction Signal: Powder Average

Powder average is the average of the diffraction pattern over all possible orientations in space with a uniform distribution. The result will be a function only of $q=\left|\underline{q}\right|$, and it will depend only on the lengths of the interatomic distances.

For a system of *N* atoms (simplified as point scatterers) with coordinates ${\underline{r}}_{j}$, j=1,…,N, each with scattering length $b_{j}$, the powder averaged intensity (differential cross section) can be written by means of the the Debye scattering equation [21] (hereafter DSE) as
(30)$$I\left(q\right)=\sum_{j,k=1}^{N}b_{j}b_{k}\mathrm{sinc}\left(2\pi q\left|{\underline{r}}_{j}-{\underline{r}}_{k}\right|\right)=\sum_{j=1}^{N}b_{j}^{2}+\sum_{j\neq k=1}^{N}b_{j}b_{k}\mathrm{sinc}\left(2\pi qd_{jk}\right)$$
with sinc(x)=sin(x)/x is the sine cardinal function and where we set $d_{jk}\equiv \left|{\underline{r}}_{j}-{\underline{r}}_{k}\right|$. For periodically ordered systems, where many of the distances $d_{jk}$ will be the same, and the scattering lengths pair is also the same. Then, we can group terms in the left sum, leaving $M_{d}$ distinct *d*-values, each with a multiplicity μ. Then, we can write
(31)$$I\left(q\right)=\;\sum_{j=1}^{N}b_{j}^{2}+\sum_{\ell =1}^{M_{d}}b_{\ell }^{2}\mu_{\ell }\mathrm{sinc}\left(2\pi qd_{\ell }\right)$$If the system is slightly disordered, the *ℓ*-indexed groups of $\mu_{\ell }$ distances might become slightly spread in value. If the spread is relatively small, we can refrain from breaking the *ℓ*-groups and instead evaluate the group average $\langle d_{\ell }\rangle$ and its variance $V_{d_{\ell }}$. Then, an effective way of modifying Equation (Equation 31) has been derived [22], with excellent approximation (see also [23,24]; this case corresponds to a paracrystalline type of disorder with no cross-interactions and with positive full correlation (value 1) along each axis. Correlation values below 1 would mean that the NC and the spacer would deform elastically to try to partially accommodate differences in size. This is a possible generalisation of this work, but we will not pursue it here as we deem it likely to be of minor importance). The modified DSE reads
(32)$$I\left(q\right)=\;\sum_{j=1}^{N}b_{j}^{2}+\sum_{\ell =1}^{M_{d}}b_{\ell }^{2}\mu_{\ell }\mathrm{sinc}\left(2\pi q\langle d_{\ell }\rangle \right)\exp\left(-2\pi^{2}q^{2}V_{d_{\ell }}\right)$$The exponential factor is the Fourier transform of a Gaussian with variance $V_{d_{\ell }}$.

We recall briefly that the DSE is just the spherical average (over all possible orientations, with uniform distribution) of the 3D scattering equation
(33)$$I\left(\underline{q}\right)=\;\sum_{j=1}^{N}b_{j}^{2}+\sum_{\ell =1}^{M_{d}^{\prime }}b_{\ell }^{2}\mu_{\ell }^{\prime }\cos\left(2\pi \underline{q}\cdot \langle {\underline{d}}_{\ell }\rangle \right)\exp\left(-2\pi^{2}\underline{q}\cdot V_{{\underline{d}}_{\ell }}\underline{q}\right)$$
where the multiplicities may differ (coincidences in 3D space are more rare). This equation is usually obtained as the square modulus of the direct Fourier transform (FT) of the scattering density:(34)$$F\left(\underline{q}\right)=\;\sum_{j=1}^{N}b_{j}\exp\left(2\pi i\underline{q}{\underline{r}}_{j}\right)$$

Suppose now that we have a distribution for 3D vector distance with a vector average and a covariance matrix (as in Equation (Equation 29)). Knowing $\langle {\underline{d}}_{\underline{K}}\rangle$ (Equation (Equation 27)) and the covariance $V_{{\underline{d}}_{\underline{K}}}$ (Equation (Equation 28)), and being
$$d_{\underline{K},\underline{g}}=\sqrt{\langle {\underline{d}}_{\underline{K},\underline{g}}\rangle \cdot \langle {\underline{d}}_{\underline{K},\underline{g}}\rangle }={\left({\left(k_{a}\langle \eta_{a}\rangle +g_{a}\right)}^{2}a^{2}+{\left(k_{b}\langle \eta_{b}\rangle +g_{b}\right)}^{2}b^{2}+{\left(k_{c}\langle \eta_{c}\rangle +g_{c}\right)}^{2}c^{2}\right)}^{1/2}$$
where the leftmost expression comes from Equation (Equation 27), we must evaluate the latter’s average and variance over the 3D distribution Equation (Equation 29).
(35)$$\begin{array}{ccc} \langle d_{\underline{K},\underline{g}}\rangle & = & \int_{R^{3}}d^{3}{\underline{d}}_{\underline{K},\underline{g}}\;P\left({\underline{d}}_{\underline{K},\underline{g}}\right)\;d_{\underline{K},\underline{g}}; \end{array}$$
(36)$$\begin{array}{ccc} V_{d_{\underline{K},\underline{g}}} & = & \int_{R^{3}}d^{3}{\underline{d}}_{\underline{K},\underline{g}}\;P\left({\underline{d}}_{\underline{K},\underline{g}}\right)\;{\left(d_{\underline{K},\underline{g}}-\langle d_{\underline{K},\underline{g}}\rangle \right)}^{2} \end{array}$$The integrals are not analytic, but a series expansion of the integrands to the second order around the averages by component of $d_{\underline{K},\underline{g}}$ yields
(37)$$\begin{array}{ccc} \langle d_{\underline{K},\underline{g}}\rangle & = & d_{\underline{K},\underline{g}}+\frac{d_{\underline{K},\underline{g}}^{2}A_{\underline{K},\underline{g}}-B_{\underline{K},\underline{g}}}{2d_{\underline{K},\underline{g}}^{3}}; \end{array}$$
(38)$$\begin{array}{ccc} V_{d_{\underline{K},\underline{g}}} & = & \frac{B_{\underline{K},\underline{g}}}{d_{\underline{K},\underline{g}}^{2}}-\frac{{\left(d_{\underline{K},\underline{g}}^{2}A_{\underline{K},\underline{g}}-B_{\underline{K},\underline{g}}\right)}^{2}}{4d_{\underline{K},\underline{g}}^{6}} \end{array}$$
where
(39)$$\begin{array}{ccc} A_{\underline{K},\underline{g}} & \equiv & \mathrm{Tr}\left(V_{{\underline{d}}_{\underline{K},\underline{g}}}\right)=\left|k_{a}\right|V_{\eta_{a}}+\left|k_{b}\right|V_{\eta_{b}}+\left|k_{c}\right|V_{\eta_{c}}; \end{array}$$
(40)$$\begin{array}{ccc} B_{\underline{K},\underline{g}} & \equiv & {\underline{d}}_{\underline{K},\underline{g}}\cdot V_{{\underline{d}}_{\underline{K},\underline{g}}}{\underline{d}}_{\underline{K},\underline{g}}={\left|k_{a}\right|}^{3}{\langle \eta_{a}\rangle }^{2}a^{2}V_{\eta_{a}}+{\left|k_{b}\right|}^{3}{\langle \eta_{b}\rangle }^{2}b^{2}V_{\eta_{b}}+{\left|k_{c}\right|}^{3}{\langle \eta_{c}\rangle }^{2}c^{2}V_{\eta_{c}} \end{array}$$We only then have to plug the $\langle d_{\underline{K},\underline{g}}\rangle$ and $V_{d_{\underline{K},\underline{g}}}$ from Equations (Equation 37) and (38) in Equation (Equation 32) in place of $\langle d_{\ell }\rangle$ and of $V_{d_{\ell }}$, respectively. The multiplicity $\mu_{\ell }$ is given in Equation (Equation 24).

## 5. Superlattice of Misaligned Objects

We also explore—partly—the case when the NCs arranged on the SL are all equal sized (no SDE) but not perfectly aligned with each other. This we name the alignment disorder effect (ADE).

We develop this case very briefly because of the extensive theoretical analysis involved that suggests dedicating a specific manuscript to it. However, we want to give at least a feeling of the effect on diffraction of alignment disorder.

Take two SL sites separated by $\underline{K}=\left(K_{a},K_{b},K_{c}\right)$ nodes, the actual displacement vector being $K_{a}{\underline{a}}_{S}+K_{b}{\underline{b}}_{S}+K_{c}{\underline{c}}_{S}$. One NC at one end of $\underline{K}$ is held fixed, an identical one at the other end is subjected to a general rotation. A general rotation in 3D space can be described as three subsequent rotations along three non-coplanar directions; for convenience, we choose the directions of ${\underline{a}}_{S},{\underline{b}}_{S},{\underline{c}}_{S}$ as axes, in the order. The rotations are quantified by three angles $\phi_{{\underline{a}}_{S}}$, $\phi_{{\underline{b}}_{S}}$, $\phi_{{\underline{c}}_{S}}$, respectively.

As for the size disorder case, we imagine an equivalent mechanism where nearest-neighbour only interactions are involved. As such, every NC has a small rotational degree of freedom with respect to its nearest neighbours. The variances of the rotation angles then increase linearly with the number of steps in each SL direction between two SL sites. It is reasonable that each SL direction influences differently the rotation angle around itself than the other SL directions. Then, we have a simple matrix equation for evaluating the angular variances:(41)$$\left(\begin{array}{c} V_{\phi_{{\underline{a}}_{S}}}\\ V_{\phi_{{\underline{b}}_{S}}}\\ V_{\phi_{{\underline{c}}_{S}}} \end{array}\right)=\left(\begin{array}{ccc} \xi & \chi & \chi \\ \chi & \xi & \chi \\ \chi & \chi & \xi \end{array}\right)\left(\begin{array}{c} K_{a}\\ K_{b}\\ K_{c} \end{array}\right)$$We require also to have no net rotation, or equivalently zero angle averages $\langle \phi_{{\underline{a}}_{S}}\rangle =\langle \phi_{{\underline{b}}_{S}}\rangle =\langle \phi_{{\underline{c}}_{S}}\rangle =0$.

The two NC spaced by $\underline{K}$ each have a diffraction amplitude $F_{NC}\left(\underline{q}\right)$ described by Equation (Equation 10). For the one NC that is rotated, $F_{NC}\left(\underline{q}\right)$ will also be rotated; we indicate it simply as $F_{NC}^{\prime }\left(\underline{q}\right)$. The total diffraction amplitude is then
(42)$$F_{NC}\left(\underline{q}\right)+\exp\left(2\pi i\underline{q}\cdot \left(K_{a}{\underline{a}}_{S}+K_{b}{\underline{b}}_{S}+K_{c}{\underline{c}}_{S}\right)\right)F_{NC}^{\prime }\left(\underline{q}\right)$$The intensity will be its square modulus
(43)$$F_{NC}^{2}\left(\underline{q}\right)+F_{NC}^{\prime 2}\left(\underline{q}\right)+2F_{NC}\left(\underline{q}\right)F_{NC}^{\prime }\left(\underline{q}\right)\cos\left(2\pi \underline{q}\cdot \left(K_{a}{\underline{a}}_{S}+K_{b}{\underline{b}}_{S}+K_{c}{\underline{c}}_{S}\right)\right)$$The term containing the product $F_{NC}\left(\underline{q}\right)F_{NC}^{\prime }\left(\underline{q}\right)$ will be greatly reduced because the rotation will cause peaks of $F_{NC}^{\prime }\left(\underline{q}\right)$ to rotate out of the corresponding peaks of $F_{NC}\left(\underline{q}\right)$ (except the origin peak, which is only relevant for SAXS, of course). The most dramatic effect will be when even the lowest lying peaks are totally decoupled. Supposing a=b=c (cubic NC cell) and $N_{a}=N_{b}=N_{C}$ (cubic NC), as the footprint of a peak in each direction extends from $-{\left(aN_{a}\right)}^{-1}$ to ${\left(aN_{a}\right)}^{-1}$, the rotation angle necessary to maximally suppress the first (100) peak located at q=1/a will be $\Phi \approx \arctan\left(2/N_{a}\right)$. This gives us a criterion for understanding when a rotation is small or disruptively large. Cumulative effects will be explored in future studies.

## 6. Example Calculations

Here, we want to show some numerical calculations of SC diffraction patterns with size disorder effect (SDE). We will start with a system that produces truly 1D scattering (a set of parallel planes does that). Then, we will have SCs with small NCs and different degrees of disorder and also different SL dimensionality (rods, planes, and true bulk SC).

### 6.1. 1D Chain of Parallel Planes with 1D Scattering

This case represents the practical case of a set of parallel planes whose diffraction is measured in $\underline{q}$-space along the direction orthogonal to the planes. This case has a great importance first of all theoretically, as sets of parallel and equispaced planes are quintessential in the definition of Bragg peaks and Bragg’s law [25]; secondly because it corresponds to lamellar order, a very common situation for soft matter systems. In fact, lamellar systems show a very similar decoherence effect, with a paracrystalline type of order range limiting effect, either stronger as in polymers with bending chains [23], with the variance of the interplanar distances proportional to the *squared* variance due likely to the concomitant effect of rotation, or weaker (the Caillé model [26,27], where the variance increases logarithmically instead because elastic deformations reduce the effect) than the one we present here (see [28] and references therein).

It is noteworthy that the dimensions orthogonal to the stacking direction (that is normal to the planes) can be ignored.

The scattering equation of this system reads
$$I\left(q\right)={\left|\sum_{m=1}^{M_{a}}\exp\left(2\pi iqz_{m}\right)\right|}^{2}=M_{a}+\sum_{m\neq m^{\prime }=1}^{N_{stack}}\cos\left(2\pi q\left|z_{m}-z_{m^{\prime }}\right|\right)$$
where $z_{m}$ are the coordinates along the stacking axis *z*, *q* is the scattering vector along the same axis, and $M_{a}$ the total number of planes. It is similar to the DSE Equation (Equation 30) where the sinc function is replaced by a more mundane cosine.

Each bunch of planes is equispaced, so we can write, for an isolated bunch of height $N_{a}$ and spacing *a*, (the latter we suppose to be the same for all bunches, the former we let be variable)
$$I_{bunch}\left(q,N_{a}\right)=\sum_{k=-N_{a}+1}^{N_{a}-1}\left(N_{a}-\left|k\right|\right)\cos\left(2\pi qka\right)$$If we average over a bunch size distribution on a finite discrete range
$$P\left(N_{a}\right),\;N_{a}=1,\ldots ,{\hat{N}}_{a}\;\left|\;\sum_{N_{a}=1}^{{\hat{N}}_{a}}P\left(N_{a}\right)=1\right.$$
with
$$\langle N_{a}\rangle =\sum_{N_{a}=1}^{{\hat{N}}_{a}}N_{a}P\left(N_{a}\right);\;V_{N_{a}}=\sum_{N_{a}=1}^{{\hat{N}}_{a}}{\left(N_{a}-\langle N_{a}\rangle \right)}^{2}P\left(N_{a}\right),$$
then we can write, for the average bunch,
(44)$$I_{bunch}\left(q\right)=\sum_{k=-\hat{N}+1}^{\hat{N}-1}\left[\sum_{N_{a}=1}^{{\hat{N}}_{a}}P\left(N_{a}\right)\max\left(0,N_{a}-\left|k\right|\right)\right]\cos\left(2\pi qka\right)\equiv \sum_{k=-\hat{N}+1}^{\hat{N}-1}\mu_{k}\cos\left(2\pi qka\right)$$

The average SL is the periodic average of the arrangement of bunches, with an average spacing $a\eta_{a}=a\left(N_{a}+s_{a}\right)$. The scattering from an SL of $M_{a}$ plane bunches then results in
$$\begin{array}{ccc} I(q) & = & \sum_{m=-M_{a}+1}^{M_{a}-1}\;\sum_{k=-{\hat{N}}_{a}+1}^{{\hat{N}}_{a}-1}\mu_{k}\;\left(M_{a}-\left|m\right|\right)\;\cos\left(2\pi qa\left|\left(k+m\left(\langle N_{a}\rangle +\langle s_{a}\rangle \right)\right)\right|\right)\times \\ & \times & \exp\left[-2m\pi^{2}q^{2}a^{2}\left(V_{N_{a}}+V_{s_{a}}\right)\right] \end{array}$$

Example 1D patterns. We consider bunches of equispaced planes (representing the NCs) stacked with dead space on top of each other, see Figure 2. The distribution is nonzero only at two values:$$\begin{array}{ccc} 1<N_{a}<5 & \to & P(N_{a})=0\\ N_{a}=5 & \to & P(5)=0.713384\\ N_{a}=6 & \to & P(6)=0.286616\\ N_{a}>6 & \to & P(N_{a})=0 \end{array}$$
resulting in
$$\begin{array}{c} \langle N_{a}\rangle =5.286616\\ V_{N_{a}}=0.204467\\ \frac{\sqrt{V_{N_{a}}}}{\langle N_{a}\rangle }=0.0855331\;\left(\approx 8.5\%\right) \end{array}$$As we see, this case results in a “narrow” distribution with 8.5% relative dispersion.

We also set a=5.431Å. We set the interbunch spacing to $0.38\langle N_{a}\rangle a$. This we suppose to have zero variance. We take $M_{a}=10$. In Figure 3, we see calculated diffraction patterns—switching on and off the 8.5% spacing dispersion

We also calculated the diffraction pattern in the case where every plane bunch (or NC) is substituted by a single scattering plane. This shows directly (Figure 4) the SL scattering and the interference (or lack thereof) when the spacing is subjected to the same 8.5% dispersion.

### 6.2. 1D, 2D and 3D SCs

We constructed cubic NCs (lattice parameter a=5 Å) with 7×7×7 unit cells (each cell containing just one point scatterer of unitary length) and arranged them on large cubic SLs with superlattice parameter $a_{S}=9.73a$. The SL dimensions were 20×1×1 unit cells (a rod-like or 1D SC), 20×20×1 unit cells (a plate-like or 2D SC), 20×20×20 unit cells (a cube-like or 3D SC). In all cases, the 1D powder diffraction trace was evaluated in a wide range with different settings of the spacing dispersion (0%, 1%, 2%, 3%, 4%, 5%). Interesting details of calculated traces are shown in Figure 5 (for the rod), Figure 6 (for the plate) and Figure 7 for the cube. Note a general reinforcement of the SL interference scattering (sharp features) as the dimensionality of the stacking increases, and also note how in general a small fractional dispersion (below 5%) is always able to destroy the SL interference on all NC Bragg peaks. In order to assess the correctness of the faster DSE computation method here exposed, we have repeated the calculations concerning Figure 6 using the simpler, much heavier but true and tested brute-force DSE evaluation. In order to do so, we have taken a 20 × 20 SL decorated with a majority of cubic NCs with six cells per side and a minority of cubic NCs with seven cells per side. This emulates a very narrow size distribution, and the chosen proportion of larger NCs is randomly assigned on the SL. The amount of larger NCs has been chosen so that the relative size dispersion amounts to 0% (ideal case), 1%, …, 5%. One hundred configurations of the SL have been generated at random, then the distances between NCs have been relaxed in order to keep a constant spacing between consecutive NCs along the SL axes. The DSE has been used then to evaluate and average over all configurations the powder diffraction patterns. The resulting plot (Figure 8) differs only slightly from Figure 6 and the residual differences can be explained with slight but unavoidable model differences.

XRPD is a simple technique and quite well suited to perform X-ray scattering experiments on SCs as a powder or floating in suspension, with real-time in-situ monitoring of the coalescence and simultaneously measuring other properties of the system. Modeling of SL disorder can be essential to reproduce observed features, as in Bertolotti et al. [15] and to correlate them to photon properties. The modeling techniques here demonstrated can be used to understand the powder diffraction signal. The information content of a 1D powder diffraction pattern is, however, limited. Other more complex and informative scattering geometries can be used to extract more information. To interpret that, it means being able to calculate model scattering in 3D reciprocal space. Every experimental geometry will explore a given subset of the full reciprocal space, so it should be easy to specialize for a given geometry. Evaluation of full 3D diffraction patterns in reciprocal space within this kind of SC modeling is the argument of the next section.

### 6.3. Atomistic Simulations and 3D Scattering

We computed the diffracted intensity (as fromEquation (Equation 34)) in the (hk0) reciprocal lattice plane of ideal and SDE-affected SLs. The ideal SL was constituted by an array of of 20×20×1 SL unit cells each “dressed” by 6×6×6 NC unit cells, here each NC and each atom of the NC sits in its ideal crystallographic position. To introduce the SDE, a fraction of 0%, 2% and 5%, of randomly chosen SL nodes was assigned to host an NC of 7×7×7 NC unit cells and then the node-to-node distance was optimised by the MC algorithm in order to maintain constant the NC spacing. The diffracted intensity shown in Figure 9 has been calculated as an average of the diffracted intensity of 100 statistically equivalent SCs “clones” generated by the MC algorithm. For these simulations, we used the same values of the SC’s and NC’s unit cells, NC spacing and atomic form factor as in Section 6.2. In Figure 9 atomic configurations obtained by MC simulations for the SDE of 0%, 2%, and 5% and the corresponding diffraction intensities in the (hk0) planes are shown. The (hk0)-patterns shown in Figure 9d,e have been calculated at the optimal sampling step to avoid finite-size contribution of the model crystal, showing horizontal and vertical diffuse scattering “strain-stripes” generated by SDE. In Figure 9g–i, the (hk0)-patterns have been calculated using a four-times finer grid to show the superposition of the domain-size with the strain effects.

## 7. Discussion

We have explored the peculiar disorder effects of NC-based SCs that constitute a growing trend because the SL order introduces or modifies physical properties in ways that are interesting for applications. To explore the underlying mechanism, a great importance is attached to the fine structural features of the superorder. As a rule of thumb, structural effects that modify the X-ray diffraction are also modifying the electronic properties through the band structure. This makes it interesting to explore the quality requirements—in terms of NC size and shape uniformity, and also in terms of co-alignment of the NCs until their periodic arrangement on an SL truly forms an SC. To this aim, we have investigated the diffraction footprint of SC whose constituting NCs have a small size dispersion that must affect the quality of the periodic SL order. It turns out that a small size dispersion (4–5%) is already able to severely affect (up to canceling) the SL coherence, whilst NC misalignment is also very effective at this task, but its destructive effect is higher for larger NC sizes. Therefore, in order to achieve SL interference effects—if they are connected to desirable changes in the electronic properties—great care must be taken to ensure a very sharp size distribution (with relative dispersion at the % order) and a great uniformity and regularity of shape (with the reasonable hypothesis that large flat NC facets would hinder misalignments).

## Figures and Tables

**Figure 1 nanomaterials-12-01781-f001:**
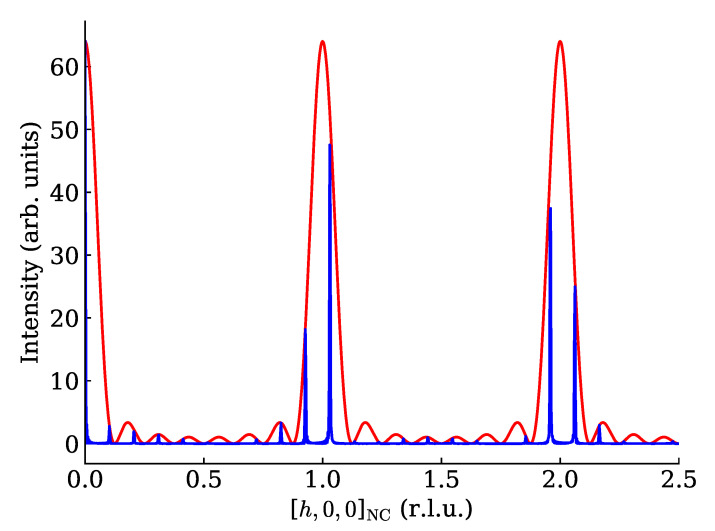
Red line: the NC scattered intensity along $\underline{q}=h{\underline{a}}^{*}$, $F_{NC}^{2}\left(h{\underline{a}}^{*}\right)$, for $N_{a}=8$. Note a sharp peak at all integer values of *h*, and the NC-suffix on the *x*-axis label indicates that the metric is defined by the NC lattice. The peaks have all the same shape and are bracketed by zeroes at $h=k\pm 1/N_{a}$, k∈Z. Factors $M_{b},M_{c}$ Blue line: $I_{SC}\left(\underline{q}\right)/M_{a}^{2}$ for $N_{a}=8$, $M_{a}=20$, ${\underline{a}}_{S}=9.7\underline{a}$. One can clearly see the very sharp SL peaks modulated by the NC scattered intensity, so that each NC peak is replaced by a tight “copse” of sharper SL peaks.

**Figure 2 nanomaterials-12-01781-f002:**
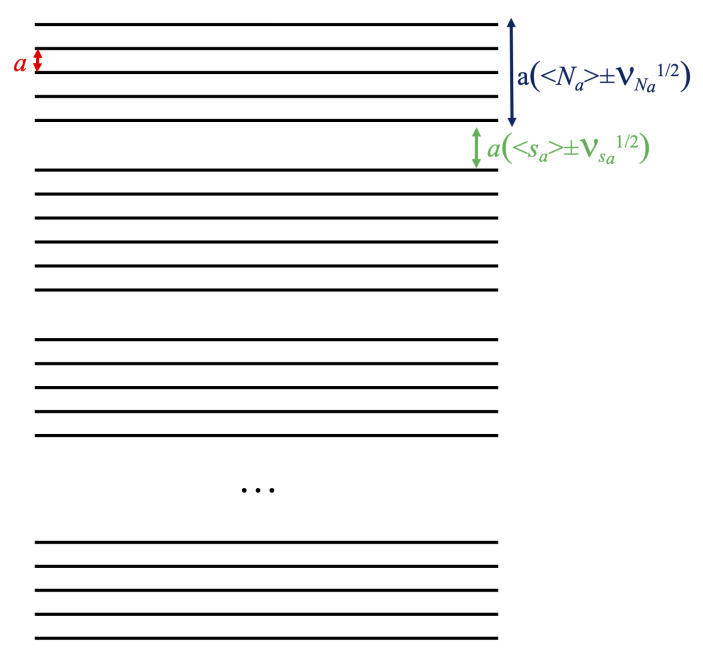
This is a representation of a system with 1D SDE. It consists of a stacking of parallel planes, sub-ordered in bunches of different height. Planes (orthogonal to the figure) are represented by their traces; the vertical direction is the stacking direction, normal to the planes. The spacing between planes in a bunch is *a*, inter-bunch spacing $a\left(\langle s_{a}\rangle \pm V_{s_{a}}^{1/2}\right)$; each bunch consists of a number of planes that on average is $\langle N_{a}\rangle$ with a dispersion $V_{N_{a}}^{1/2}$.

**Figure 3 nanomaterials-12-01781-f003:**
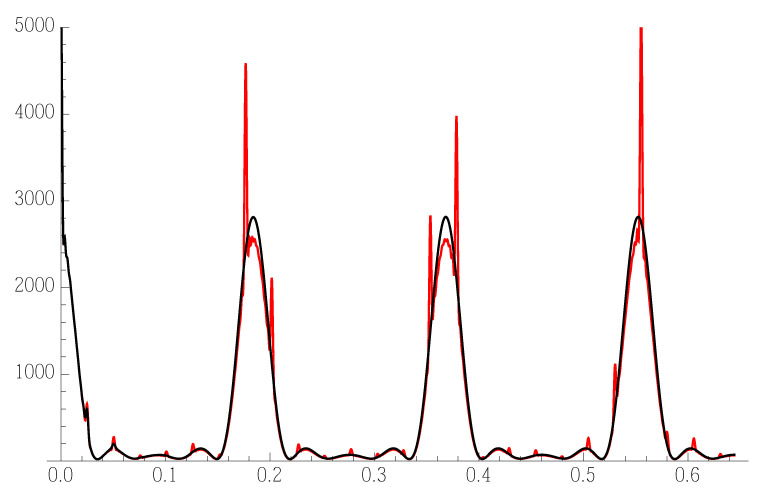
Plane bunch sequence 1D diffraction pattern: Red—calculation with zero spacing dispersion; Black—including 8.5% spacing dispersion as from the text. The 8.5% dispersion destroys the sharp small peaks (SL interference) except in the small-angle region.

**Figure 4 nanomaterials-12-01781-f004:**
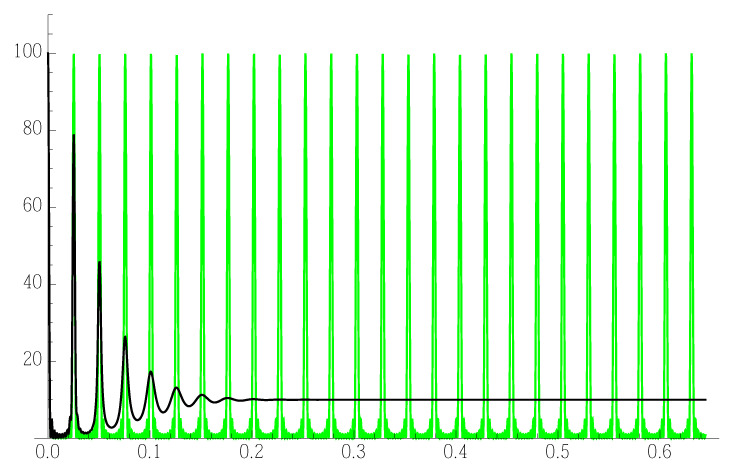
Basic 1D diffraction pattern of the same sequence, just substituting the plane bunches with single plane scatterers—to see the naked decoherence effect of the spacing dispersion. Green—zero dispersion, black—8.5% dispersion.

**Figure 5 nanomaterials-12-01781-f005:**
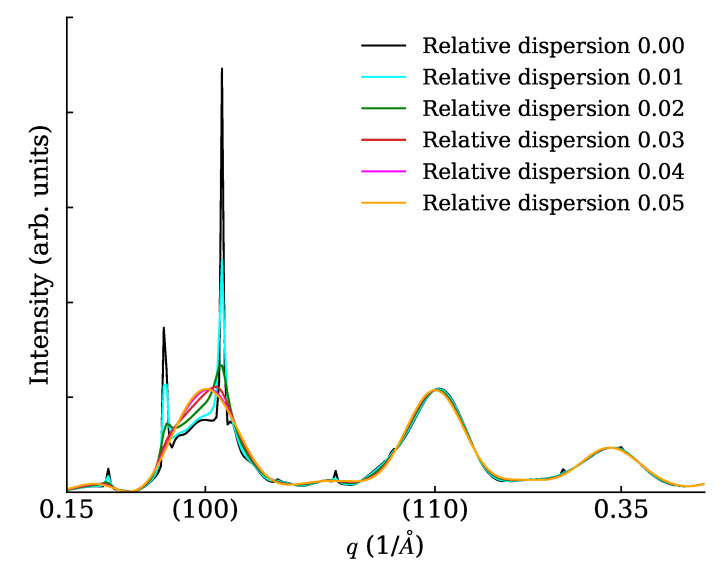
XRPD scattering from a rod (1D SC) with various levels of relative SL spacing dispersion.

**Figure 6 nanomaterials-12-01781-f006:**
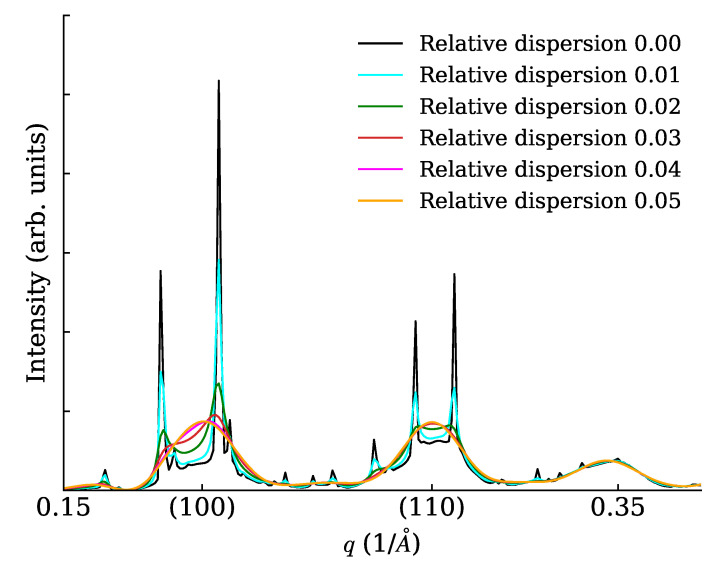
XRPD scattering from a plate (2D SC) with various levels of relative SL spacing dispersion.

**Figure 7 nanomaterials-12-01781-f007:**
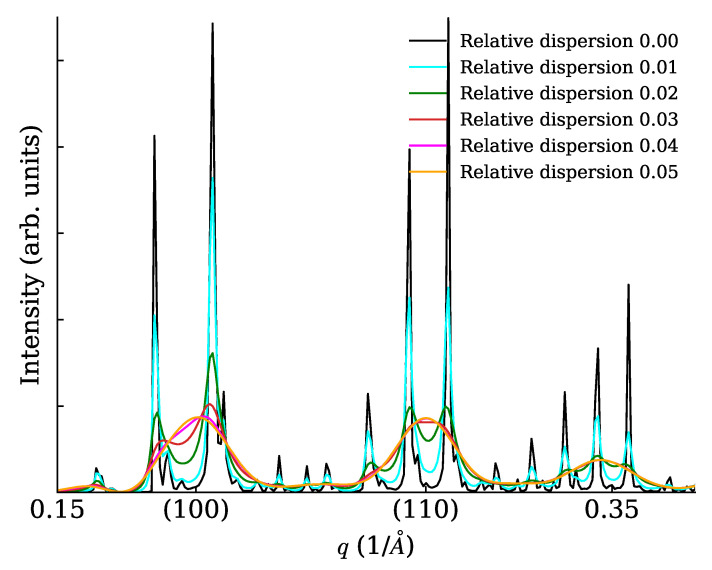
XRPD scattering from a cube (3D SC) with various levels of relative SL spacing dispersion.

**Figure 8 nanomaterials-12-01781-f008:**
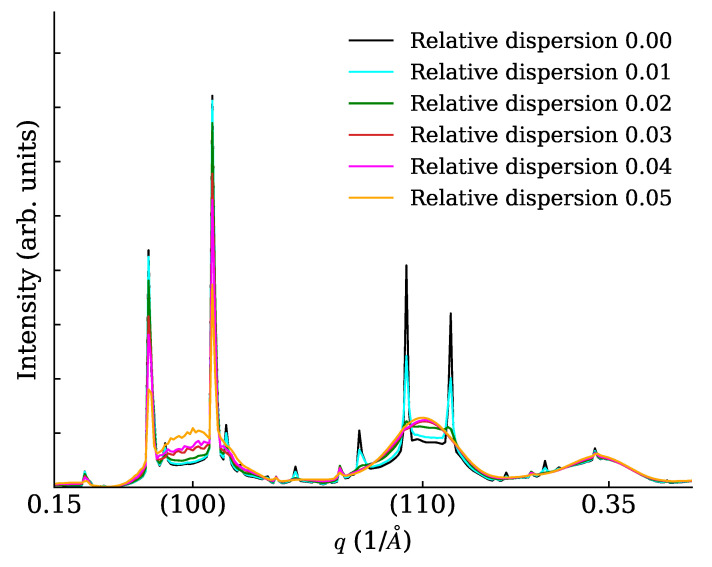
XRPD scattering from a plate (2D SC) with various levels of relative SL spacing dispersion, approximately the same as in Figure 6 but evaluated by brute force.

**Figure 9 nanomaterials-12-01781-f009:**
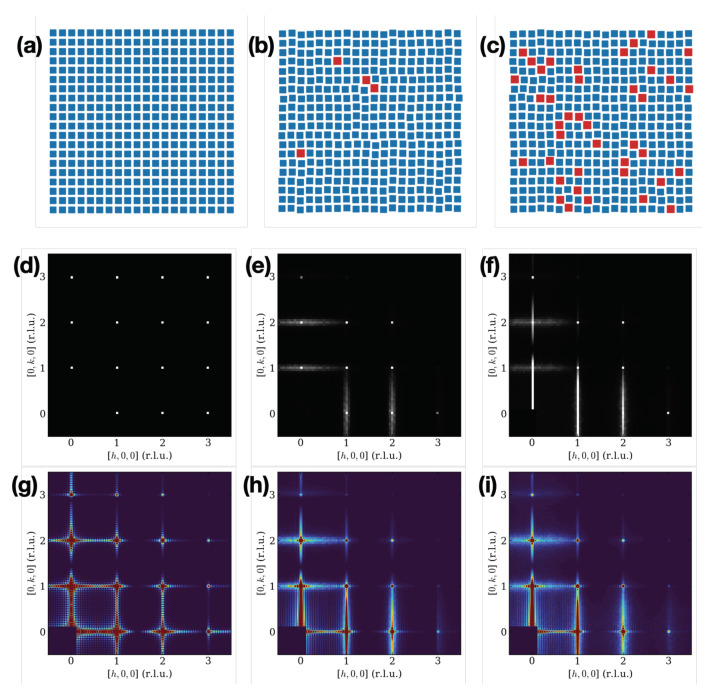
Top row: atomic configurations 20×20×1 SLs with variable degree of SDE: (**a**) 0%, (**b**) 2%, and (**c**) 5%. Red squares indicate sites occupied by the larger NC size. Middle row: (hk0) reciprocal lattice planes for the degree of SDE: (**d**) 0%, (**e**) 2%, and (**f**) 5%. The diffracted intensities were calculated using a sampling step of 1/20 SC’s unit cells to avoid finite size-effects contributions to the calculated intensities. The FT of the simulated crystal can be written as $F\left(\underline{q}\right)=F_{\infty }\left(\underline{q}\right)⨂\mathrm{Box}\left(\underline{q}\right)$, where $F_{\infty }\left(\underline{q}\right)$ is the structure factor of the infinite crystal, and $\mathrm{Box}(\underline{q})$ the FT of the shape function of the finite crystal. This shows that it is possible to avoid the contribution fo limited size effects by calculating the diffracted intensity where the function $\mathrm{Box}(\underline{q})\equiv 0$. Bottom row: (hk0) reciprocal lattice planes for the degree of SDE: (**g**) 0%, (**h**) 2%, and (**i**) 5%. The diffracted intensities were calculated using a sampling step of 1/80 SC’s unit cells to display finite size-effects contributions to the calculated intensities. All intensities are in electron units and the color maps scale is fixed to 10^−2^ of the strongest Bragg peak to make diffuse effects visible. Origin (000) peak purposely masked.

## Data Availability

Not applicable.

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
