# Peer review of "Diffraction from Nanocrystal Superlattices"

_nanomaterials, 2022, doi:10.3390/nano12101781_

Round 1
Reviewer 1 Report
This manuscript presents methods to calculate the diffraction from supercrystals.
Different situations are examined. First (section 3), the ‘ideal’ situation of a superlattice of identical nanocrystals is described. Then two more realistic situations are considered for two different types of disorder: size disorder effect (section 4) and orientational disorder of the nanocrystals (section 5).
The content of this manuscript will be very useful for the increasing community of researchers studying supercrystals using X-ray scattering. I recommend the publication of this work with the minor corrections listed hereafter :
- Figure 4 corresponds to a subcase of 1D ordering where each nanocrystal is replaced by a single scattering plane. The authors should mention that this subcase is a very important one because it corresponds to lamellar order, a very common situation for soft matter systems. The results should be compared to the abundant associated literature about the scattering by lamellar phases. See for example the recent tutorial review by Ian Hamley (Soft Matter, 2022, 18, 711–721 | 711, DOI: 10.1039/d1sm01758f) and the cited references.
- In Figure 2, please correct the text written in green; it should correspond to sa.
- In the subsections 6.2 and 6.3, more text is needed to explain how the presented calculations are illustrating general rules on the effect of disorder.
- In section 6.3, it is briefly mentioned that the nanocrystals are made of Ni (Nickel) atoms (line 128). Is it really necessary? A better option is probably to stay very general as in the other sections. Again, section 6.3 needs more explanations and context.
- Some typos need to be corrected.
Reviewer 2 Report
The authors discuss the effects of nanocrystals’ size dispersion and orientation disorder to powder diffraction profiles from supercrystals (i.e., assembly of nanocrystals). In particular, they derived a detailed analytical treatment to model the diffraction profiles of simple super lattice, yet some aspects are postponed to future work publication. The mathematical treatment is rigorous and of potential interest.
Discussions and results are, however, difficult to understand from a broader audience than experts in the specific mathematical solution of scattering data. The authors do not provide relevant validation of the results: not against experimental evidence examples (their own or from available literature) nor against full scale simulations (e.g., via brute force solution of the DSE). Almost the entire manuscript is dedicated to report of mathematical formulations and plots of the corresponding predicted scattering profiles. Possible errors (if any) and effect of approximations are not assessed.
In the opinion of this reviewer the presentation of the results and the discussion of how these fits within available literature have to be significantly improved before the manuscript is suitable for publication.
See the attached pdf for more details.

Round 2
Reviewer 2 Report
The manuscript quality and completeness have improved significantly. I believe the manuscript is now suitable for publication.